# Responses of Bilevel Ventilators to Unintentional Leak: A Bench Study

**DOI:** 10.3390/healthcare10122416

**Published:** 2022-11-30

**Authors:** Marius Lebret, Emeline Fresnel, Nathan Prouvez, Kaixian Zhu, Adrien Kerfourn, Jean-Christophe Richard, Maxime Patout

**Affiliations:** 1Med2 Lab, Air Liquide Medical Systems, 92022 Antony, France; 2Erphan, Paris-Saclay University, UVSQ, 78646 Versailles, France; 3KerNel Biomedical, 76100 Rouen, France; 4Vent’ Lab, Medical Intensive Care Unit, Angers University Hospital, 49000 Angers, France; 5Service des Pathologies du Sommeil (Département R3S), AP-HP, Groupe Hospitalier Universitaire APHP-Sorbonne Université, site Pitié-Salpêtrière, F-75013 Paris, France; 6INSERM, UMRS1158 Neurophysiologie Respiratoire Expérimentale et Clinique, Sorbonne Université, F-75005 Paris, France

**Keywords:** noninvasive ventilation, mask leak, air leak, bench study, chronic respiratory failure

## Abstract

Background: The impact of leaks has mainly been assessed in bench models using continuous leak patterns which did not reflect real-life leakage. We aimed to assess the impact of the pattern and intensity of unintentional leakage (UL) using several respiratory models. Methods: An active artificial lung (ASL 5000) was connected to three bilevel-ventilators set in pressure mode; the experiments were carried out with three lung mechanics (COPD, OHS and NMD) with and without upper airway obstruction. Triggering delay, work of breathing, pressure rise time, inspiratory pressure, tidal volume, cycling delay and the asynchrony index were measured at 0, 6, 24 and 36 L/min of UL. We generated continuous and inspiratory UL. Results: Compared to 0 L/min of UL, triggering delays were significantly higher with 36 L/min of UL (+27 ms) and pressure rise times were longer (+71 ms). Cycling delays increased from −4 [−250–169] ms to 150 [−173–207] ms at, respectively 0 L/min and 36 L/min of UL and work of breathing increased from 0.15 [0.12–0.29] J/L to 0.19 [0.16–0.36] J/L. Inspiratory leakage pattern significantly increased triggering delays (+35 ms) and cycling delays (+263 ms) but decreased delivered pressure (−0.94 cmH_2_O) compared to continuous leakage pattern. Simulated upper airway obstruction significantly increased triggering delay (+199 ms), cycling delays (+371 ms), and decreased tidal volume (−407 mL) and pressure rise times (−56 ms). Conclusions: The pattern of leakage impacted more the device performances than the magnitude of the leakage per se. Flow limitation negatively reduced all ventilator performances.

## 1. Introduction

Non-invasive positive-pressure ventilation (NIV) is the first choice of long-term, home treatment for chronic hypercapnic respiratory failure (CH-RF) [1]. The use of long-term NIV has tremendously increased over the last 2 decades in most countries, in particular for the treatment of Chronic Obstructive Pulmonary Disease (COPD), Obesity Hypoventilation Syndrome (OHS) and Neuromuscular Diseases (NMD) [2,3]. Data have shown that NIV improves both physiological and clinical outcomes in all precited diseases [4,5,6].

Unintentional leaks can occur during long-term NIV. Unintentional Leaks can alter NIV clinical efficacy and decrease NIV tolerance [7,8]. Leaks can also impair devices performances [9,10,11]. Hence, nowadays, clinicians monitor unintentional leaks while initiating NIV or during follow-up [11,12,13].

Monitoring NIV has become easier thanks to the development of remote monitoring [14,15]. Apart from daily use of the ventilator, all monitored data (triggered breaths, residual apnea hypopnea index, estimated volume…) are derived from the flow delivered by the ventilator. As leaks lead to an increase of the ventilator flow, the reliability of monitored data may be reduced in the presence of leaks [16]. However, to date, the impact of leaks has mainly been assessed in bench models that did not reflect leaks as they occur in real-life or only in ventilators used for patients requiring life-support ventilator. Indeed, in real-life most leaks are unintentional and not predictable [17,18,19]. Hence, they may have variable intensity and different patterns: inspiratory or continuous leakage, mouth leak, leaks created by sleep position change or device pressure change, transient leakage owing to mask displacement, etc. [20].

Our hypothesis was that non-continuous leakage had more detrimental impact on ventilator performances than continuous leakage, particularly regarding triggering and cycling delays. Therefore, our aim was to assess the impact of the level and pattern of unintentional leakage as well as the influence of different respiratory models on home bilevel ventilators performances and synchrony in the most used ventilators currently available.

## 2. Materials and Methods

Our bench model consisted of an active artificial lung model (ASL-5000, Ingmar Medical, Pittsburgh, PA, USA) connected to three bilevel ventilators: Vendom 40 (Air Liquide Medical Systems, Antony, France), Lumis 150 VPAP ST (Resmed, San Diego, CA, USA) and DreamStation BIPAP S/T (Philips Respironics, Murrysville, PA, USA); respectively named V40, L150 and DS thereafter. We used a 22 mm single limb circuit of 1.8 m long for all experiments. Respiratory efforts were simulated with a muscular pressure curve model depending on two parameters which are the airway pressure drop at 100 ms (P0.1) and the respiratory rate (RR); see Fresnel et al. [21] work for detailed respiratory effort settings used in this bench study. This effort was combined with three different lung mechanics conditions, reflecting the pulmonary function of the simulated patients by modulating resistance (R) and compliance (C) parameters; 20 respiratory cycles were simulated for each condition.

### 2.1. Simulated Airway Obstruction

For each respiratory model we simulated upper airway obstruction by increasing the resistance parameter in the ASL 5000 settings. This created a flow limitation in addition to the initial parameters of each model, mimicking situations seen when obstructive respiratory events appear during sleep. All the simulation parameters are reported Table 1.

### 2.2. Intentional and Unintentional Leaks

We placed a T-piece between the ASL 5000 and the ventilator circuit. T-piece was connected to a calibrated intentional leak port fixed at 34 l/min at 10 cmH_2_O (corresponding to the median intentional leak of the principal facial masks available on the market nowadays).

We generated two types of unintentional leaks: continuous and inspiratory leaks. To simulate various rates of unintentional leakage we used a variable opening valve downstream to the calibrated intentional leak port. In case of inspiratory leaks, we added a Threshold PEP device set at 6 cmH_2_O (Philips Respironics, Murrysville, PA, USA) allowing leaks to only occur during insufflation when the pressure in the circuit was higher than the end positive expiratory pressure set. Leak patterns are likely to affect device trigger regulations, yet the influence of leakage on device performances is usually assessed using continuous (linear) models of leakage [22] which is not representative of that occurs in real life owing to several reasons such as: mouth breathing, patient position, phase of the respiratory cycle or respiratory events [20].

Total leak was continuously monitored using a pneumotachograph (SFM3000, Sensirion, Stäfa, Switzerland). Each level of unintentional leakage was systematically titrated at 10 cmH_2_O before each experiment. Figure 1 depicts the experimental model.

### 2.3. Ventilator Settings

The devices were set in pressure support mode and all experiments were run without humidifiers. Because expiratory trigger sensitivity and pressure rise time are graduated in different units in each device, these settings were comparatively evaluated using the ASL-5000 to establish equivalent sensitivities and rise time among devices. Detailed settings regarding expiratory trigger and pressure rise time for each device is presented in Appendix A. Ventilators were set as follows: inspiratory pressure: 15 cmH_2_O, expiratory pressure: 5 cmH_2_O, inspiratory time window: 0.5–1.5 s (when available), backup rate: 7 cycles per minute (or turned off when possible). At the onset of each experiment, the inspiratory trigger sensitivity was set at the intermediate value for L150 and V40 (and in Autotrak mode for DS) as previously described by Zhu et al. [19].

### 2.4. Ventilator Performance and Synchrony Indicators

To evaluate our outcomes, we computed the following performance indicators of ventilators:(1)Triggering delay (or Time to trigger) in ms, by measuring the time lag between the onset of the simulated effort and the onset of the pressure support;(2)Work of breathing (or WOB) in J/s, computed as the integral of the product of the muscular pressure and the flow during the inspiratory phase and reported to the tidal volume;(3)Pressure rise time in ms, defined as the time required to reach the set pressure during the inspiratory phase;(4)Delivered inspiratory pressure in cmH_2_O, defined as the peak pressure reached during the inspiratory pressurization phase;(5)Tidal volume in ml, defined as the difference between the maximal volume delivered within the current cycle to the mechanical lung and its residual volume;(6)Cycling delay in ms, by measuring the time lag between the expiratory pressure release and the end of the patient’s neural inspiration.

Description of how ventilator performance was assessed is depicted in Appendix A.

Patient-ventilator asynchrony is defined as the mismatching between neural and mechanical inspiratory time [23,24] and can be assessed using the asynchrony index. According to the framework proposed by the SomnoNIV group, the following “simulated patient” ventilator asynchrony (sPVA) events were assessed [13]:Ineffective efforts (IE) characterized by an inspiratory effort not assisted by the ventilator. It can be identified as a drop of airway pressure associated with an increase or decrease of airflow (if occurring during expiratory or inspiratory phase, respectively).Auto-triggering (Auto), characterized by the presence of mechanical cycles unrelated to the patient’s spontaneous breathing.

The asynchrony index (AI) provided as a percentage (%), was calculated as the total number of asynchronous cycles (ineffectively triggered breaths plus auto-triggered breaths) divided by the number of simulated respiratory cycles for each experiment.

### 2.5. Experimental Protocol

Experiments started with 0 l/min of unintentional leak and lasted 20 respiratory cycles. At the end of these 20 cycles, if the simulated patient-ventilator asynchrony (sPVA) rate was inferior to 25% [19], the unintentional leak would be manually increased in a stepwise manner as follows: 6, 24 and 36 l/min. If sPVA were equal of superior to 25%, then the last level of unintentional leak was maintained while the inspiratory trigger was adjusted in a stepwise manner to decrease the sPVA rate as follows: increase the inspiratory trigger sensitivity if predominant sPVA were ineffective efforts or decrease of the inspiratory trigger sensitivity if predominant sPVA were auto-triggered cycles. This procedure was repeated until the critical leak was reached or until 36 l/min, the maximum leak level initially planned in our experimental protocol. An “effective” adjustment was defined as a manual adjustment of trigger sensitivity that decreased sPVA below 25%. The sPVA threshold of 25% was based on a previous work published by Zhu et al. [19]. This procedure was repeated for each ventilator crossed with the three respiratory conditions (COPD, OHS and NMD), with continuous and inspiratory unintentional leak and last with and without simulated upper airway obstruction.

### 2.6. Statistical Analysis

Results are expressed as median and interquartile intervals. Mann–Whitney or Kruskal–Wallis tests were used to compare independent continuous variables. Wilcoxon or Friedman tests were used to compare continuous dependent variables. In that case, we used the setup without unintentional leakage as reference and applied Dunn’s correction for multiple comparisons. All tests were two-sided. For all tests, significance level was set at 0.05. Statistical analysis was performed using Prism 9.0.0 (GraphPad Software, La Jolla, CA, USA).

## 3. Results

Influence of leakage rate

Triggering delays were preserved until 24 l/min of unintentional leakage but were significantly higher with 36 l/min of unintentional leakage (+27 ms with 36 l/min of unintentional leak compared to the absence of unintentional leakage, *p* < 0.01). Pressure rise times were longer at 36 l/min (+71 ms, *p* > 0.001 compared to the absence of unintentional leakage). Cycling delays were affected by leakage greater or equal to 24 l/min: they increased from −4 [−250–169] ms without unintentional leak to 150 [−173–207] ms with 36 l/min of unintentional leak (*p* < 0.002). Work of breathing significantly increased with higher unintentional leak: from 0.15 [0.12–0.29] J/l without unintentional leak to 0.19 [0.16–0.36] J/l with 36 l/min of unintentional leak (*p* < 0.001). Delivered pressure was maintained for all leak flow rates above the set IPAP and no significant difference was observed for delivered volume (Figure 2).

Influence of leak pattern

Continuous leakage pattern had a similar impact on triggering delay, delivered pressure, and cycling delay than the control situation (ie. no leakage) (Figure 3). Inspiratory leakage pattern increased triggering delays (+35 ms, *p* = 0.002), delivered pressure was lower (−0.94 cmH_2_O, *p* < 0.001) and cycling delays were longer (+263 ms, *p* < 0.001) compared to continuous leakage pattern. Works of breathing, pressure rise times and delivered volumes were not significantly affected.

Influence of simulated upper airway obstruction

Triggering delay significantly increased for all devices in the case of a simulated upper airway obstruction (+199 ms, *p* < 0.001) while tidal volume was significantly decreased (−407 mL, *p* < 0.001). The simulation of upper airway obstruction moderately shortened pressure rise times (−56 ms, *p* < 0.001) but significantly increased cycling delays (+371 ms, *p* < 0.001). Work of breathing was not significantly impacted (*p* = 0.34) (Figure 4).

Influence of respiratory models

In COPD model triggering and cycling delays were increased with respect to OHS model (respectively +52 ms, *p* < 0.009 and +264 ms, *p* < 0.02), cycling delay was also augmented with COPD model compared to NMD model (+256 ms, *p* < 0.0001). Work of breathing was significantly higher in OHS model (+0.07 J/L vs. COPD model, *p* < 0.0002 and +0.12 J/L vs. NMD model, *p* < 0.0001). Rise time was higher in NMD compared to OHS and COPD model (respectively +63 ms, *p* < 0.009 and +76 ms, *p* < 0.014). Last, tidal volume was highly augmented with NMD (+285 mL vs. COPD, *p* < 0.0001 model and +323 mL vs. OHS model, *p* < 0.0001) (Appendix A).

Impact of leak patterns on asynchrony index

As shown in Figure 5, the gradual increase in unintentional leak rate did not significantly affect the occurrence of patient-ventilator asynchronies when the leak pattern was continuous. Conversely, inspiratory leak pattern significantly increased PVA rate at 36 L/min (*p* = 0.04) compared with lower leak flow rates.

Difference between ventilators

Regarding the device used, no significant difference was observed between the devices for triggering delays, cycling delays, or delivered volume (*p* = 0.35, *p* = 0.9 and *p* = 0.7, respectively). Work of breathing was significantly increased with DS (*p* = 0.05) and IPAP was always maintained for all devices above the set IPAP, with higher pressures observed with L150 (*p* = 0.012) (Appendix A).

## 4. Discussion

In this bench study, we demonstrated that the pattern of leakage (intermittent leaks) had more impact on the device performances and the patient-ventilator asynchronies than the magnitude of the leak flow rate itself. We also evidenced that flow limitation negatively impacted all performance indicators.

Unintentional leakage is the main adverse effect encountered in the setting of long-term ventilation [9,11,12,25]. Several strategies can be proposed to reduce leakage such as mask size change, a careful choice of the type of interface, switching from a nasal to an oronasal mask when mouth leaks are suspected [26,27], or carefully reducing the pressure [26,28]. Unfortunately, such interventions are not always sufficient to tackle recalcitrant unintentional leakage since several determinants of unintentional leak exist and may be involved [26]. If leakages can reduce patients’ comfort and observance over the long term [29,30], our results demonstrate that bilevel ventilators can nevertheless support high unintentional leak rate without degrading their performances nor creating PVA until a certain level of leak flow rate is reached. To date, no clear cut-off for the acceptable level of unintentional leakages has been determined. A leak threshold of 24 l/min [31] is often considered as the acceptable limit beyond which a clinical intervention is required, although this limit is not based upon evidence [27,30]. This threshold is probably clinically irrelevant since our results demonstrated that ventilator performances and synchronization were not only determined by leakage rate, but also linked to (i) the leak pattern and (ii) the presence of flow limitation. This is of great importance while adjusting NIV parameters, since special attention should be paid to the nature of leakage, e.g., sawtooth leaks created by recurrent mouth openings, even at low leak flow rates, which could strongly impact the performance of the device and its synchronization. Hence, clinicians (physicians, physiotherapists, nurses, etc.) should aim at reducing unintentional leakage as much as possible, even at low leakage rate, in order to tackle potential NIV-related side effects that are particularly common [32]. In addition, the clinical consequences of leakage should not be forgotten. That is why it is important to ask the patient directly about the potential adverse effects of leakage, even if the leakage reported by the telemonitoring system seems low.

Interestingly, performances indicators did not differ when comparing consolidated continuous leakages (Figure 2) to the control situation (absence of unintentional leak) suggesting that device algorithms can cope with leaks when they are constant but have difficulties as soon as the leak is polymorphic or not homogeneous.

Although the algorithms of the ventilators differ from each other, some general hypothesis may explain why the intermittent leakages may have a more detrimental impact than continuous leakages on device synchronization.

(i)Leak compensation algorithms probably calculate the mean unintentional leakage rate. if the leakage appears only during the inspiratory phase, thus the mean leakage rate is likely to be underestimated assuming a situation of continuous leakage (expiratory and inspiratory leak). In these conditions, the device may interpret the unintentional leak as patient flow, which leads to cycling delays since the patient flow drop is not correctly distinguished from the leak flow;(ii)On the other hand, during exhalation, the unintentional leak is probably overestimated since the averaging considers the inspiration additional leak. This may lead to a decreased sensitivity of the inspiratory trigger thresholds.(iii)Last, during the conception phase, ventilator devices are designed and tested using calibrated leak port only, providing continuous and rather stable leakage rates.

In our study, the type of ventilator had little influence on the occurrence of PVA. To a certain extent they all failed to maintain PVA rate under 25% in the same conditions. We naturally observed some discrepancy between devices, but the main drivers were still the pattern of the leak, the presence of an upper airway flow limitation (simulated by the bench) and the mechanical property of the models.

Contal et al. demonstrated that the algorithms of the different devices differ when it comes to leakage estimation [16]. Nevertheless, the purpose of our study was not to assess the performance of the devices in estimating leakage; Zhu et al. already showed the great difference between actual level of leak and what is estimated by the devices [19]. As with CPAP devices, bilevel ventilators do not report leak rate the same way among manufacturers, which is detrimental when monitoring NIV. For example, some manufacturers report global leaks while others report only estimated unintentional leaks. Some report mean or median leaks, 90th percentile or 95th percentile. This makes it difficult for medical staff to read the devices reports, creates confusion and makes it difficult to change ventilators for the same patient, when necessary, since the leaks data cannot be compared. We suggest, as many other authors, that leakage (unintentional and intentional) should be reported in a standardized manner by all the manufacturers, which would greatly facilitate patient’s monitoring, as well as research regarding leakage influence on clinical outcomes and technical performance of the devices.

The main limit of our work was that we simulated the upper airway flow limitation using the ASL 5000, whereas one can argue that using a starling resistance system to mimic upper airway closure is more realistic. We nevertheless obtained similar results than Zhu et al. when flow limitation was applied, which makes us think that our flow limitation model was acceptable. Second, we only tested two type of leak patterns, one continuous and one inspiratory only. It would have been relevant to evaluate a random anarchic pattern of unintentional leakage or expiratory leak only, as made by Sogo, Luján et al. [33,34,35,36]. It is likely that the presence of unintentional random anarchic leakage patterns can impact the functioning of the machines even more, since the irregular character of the leakage flow is not well corrected by the algorithms. We are currently working on the simulation of random patterns of anarchic leakage, which could allow in the future to evaluate more accurately the behavior of the devices.

## 5. Conclusions

In summary, this bench highlighted the good performance of bilevel ventilators in the presence of high-level unintentional leakage. These results also suggest that home bilevel ventilators are as effective in managing leakage than life supports [19]. We also contributed to demonstrate that there is no generic threshold of “critical leak” and that the intermittent, or polymorphic nature of unintentional leakage has a more deleterious impact on device performance and synchronization, than the leak flow rate of continuous leakage. Bench studies evaluating noninvasive ventilation devices should include not only an unintentional leak port, but also allows for creating polymorphic or dynamic leak patterns. Last, in a context of democratization of remote monitoring, it is thus advisable to be particularly attentive in case of high unintentional leakage, heterogeneous leakage or important respiratory flow limitation. Indeed, it is very likely that this can degrade the quality of the ventilation, but also the quality of the data estimated/measured by the devices and transmitted to the physician and the home care provider.

## Figures and Tables

**Figure 1 healthcare-10-02416-f001:**
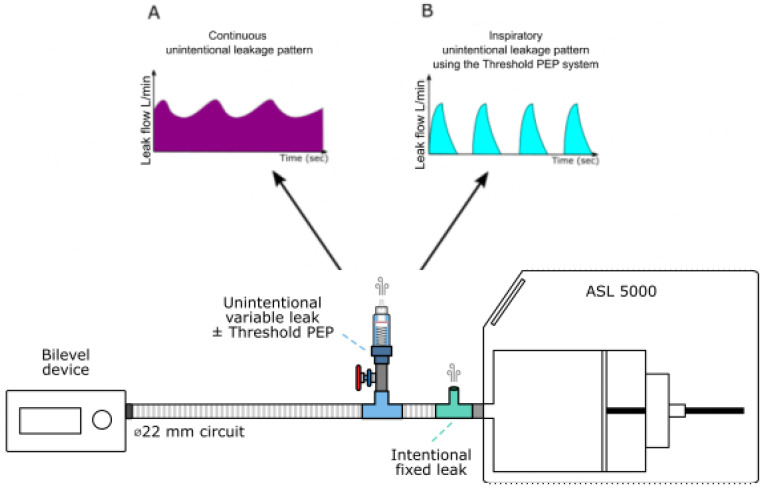
Experimental setup used to generate intentional and unintentional leakages. ASL 5000: artificial lung. (**A**) represents a schematization of the continuous leakage pattern observed when the Threshold PEP system is not activated. (**B**) represents a schematization of the inspiratory unintentional leakage (intermittent) pattern observed when the Threshold PEP is activated: in that condition, leakage occurs only when the pressure is higher or equal to 6 cmH20, therefore only during the inspiratory phase.

**Figure 2 healthcare-10-02416-f002:**
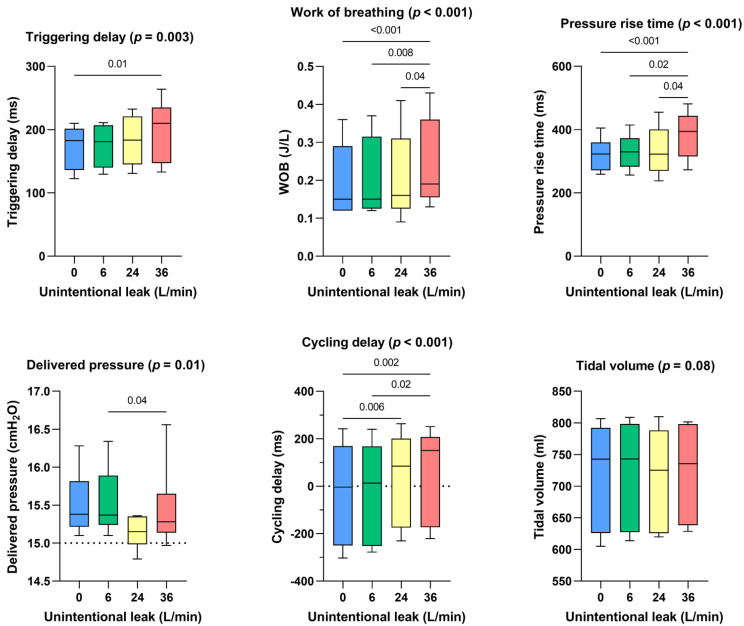
Consolidated data integrating the 3 respiratory models/devices without flow limitation, for the dose response effect between the level of continuous unintentional leakage (0, 6, 24 and 36 L/min) and the following indicators: triggering delays, work of breathing, pressure rise time, delivered pressure, cycling delay and tidal volume.

**Figure 3 healthcare-10-02416-f003:**
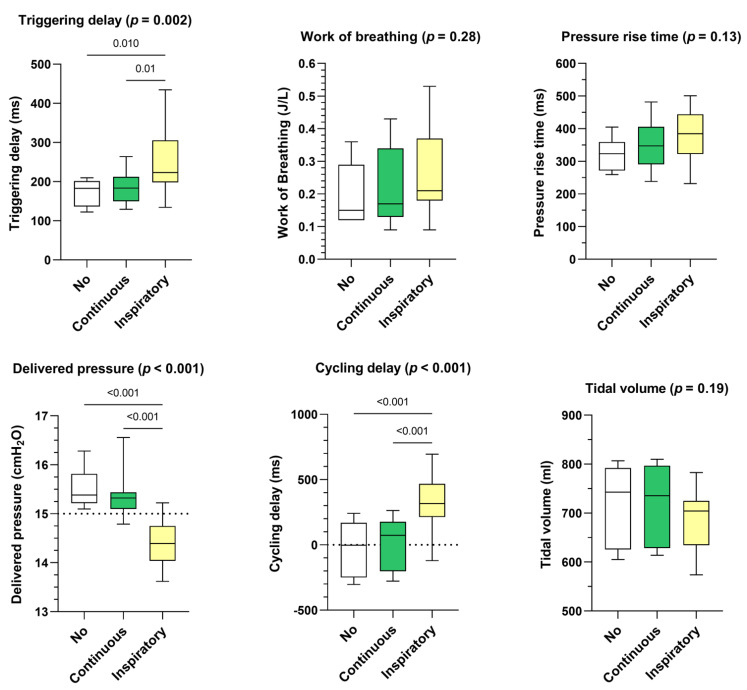
Consolidated data integrating the 3 respiratory models/devices without flow limitation according to the pattern of unintentional leak (continuous versus inspiratory) on triggering delays, work of breathing, pressure rise time, delivered pressure, cycling delay and tidal volume. Box plots «No» represent the control situation, i.e., without unintentional leakage.

**Figure 4 healthcare-10-02416-f004:**
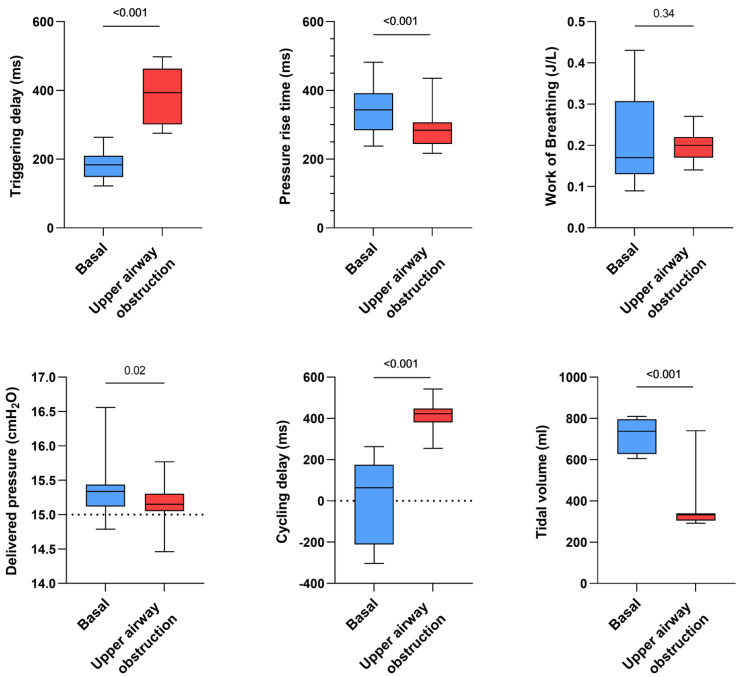
Consolidated data integrating the 3 respiratory models/devices and the 3 ventilators with continuous and inspiratory leakage according to airway patency status (upper airway obstruction versus absence of upper airway obstruction, both simulated by increasing or not the resistance settings on the ALS 5000) on triggering delays, work of breathing, pressure rise time, delivered pressure, cycling delay and tidal volume.

**Figure 5 healthcare-10-02416-f005:**
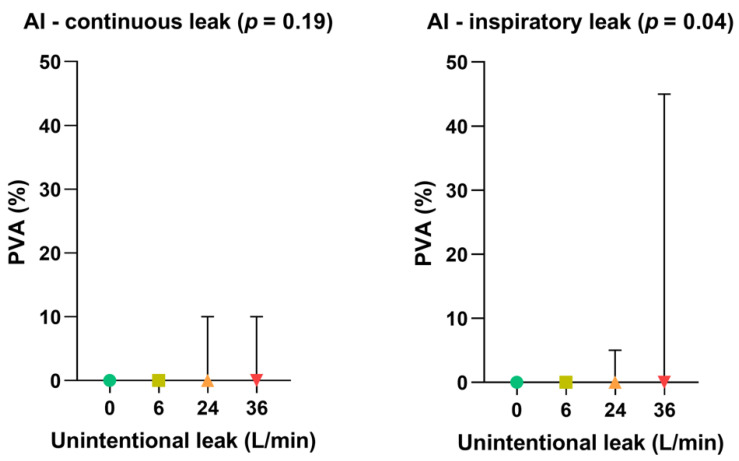
Consolidated data integrating the 3 respiratory models/devices without flow limitation according to the pattern of unintentional leak (continuous versus inspiratory) on patient-ventilator asynchrony (PVA) rates. Symbols and associated bars represent the median values and interquartile ranges. Liter per minute (L/min): Asynchrony Index (AI).

**Table 1 healthcare-10-02416-t001:** Parameters used to simulate pulmonary mechanics and respiratory dynamics with the mechanical lung.

Model	Simulated Airway Obstruction (Flow Limitation)	Inspiratory Resistance (cmH_2_O/L.s)	Expiratory Resistance (cmH_2_O/L.s)	Compliance (mL/cmH_2_O)	P0.1 (Pmax) (cmH_2_O)	RR (Breaths/min)
OHS	no	8	5	30	3 [12]	15
	yes	40				
COPD	no	20	25	50	3 [14]	12
	yes	60				
NMD	no	5		60	1 [5]	20
	yes	25				

## Data Availability

Data are available upon request to the corresponding author.

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
