# Peer review of "Responses of Bilevel Ventilators to Unintentional Leak: A Bench Study"

_healthcare, 2022, doi:10.3390/healthcare10122416_

Round 1

Reviewer 1 Report

thanks for the opportunity to revise this manuscript. It is a bench study which focused on studying the effect of either continuous or intermittent non intentional leaks on three different ventilator .

the study is very interesting I have a few comments that I would like to be more specified:

1. method :why the non intentional leakage port was positioned between the ventilator and the intentional leakage port? Considering that the usual reasons for non intentional leaks are "mouth breathing, patient position, phase of the respiratory cycle or respiratory events" as per author description, wouldn't be more appropriate to  positioning the  non intentional leaks port before the intentional leak port?

2. line 242 please delete the sentence which I guess belongs to the word basic journal format

3. discussion: among the several options discussed I would suggest to add that to improve intentional leaks, also the careful choice of which interface to use can guarantee the NIV success ref: 10.1080/17476348.2022.2121706

Author Response

thanks for the opportunity to revise this manuscript. It is a bench study which focused on studying the effect of either continuous or intermittent non intentional leaks on three different ventilator .

the study is very interesting I have a few comments that I would like to be more specified:

  1. methods:why the non intentional leakage port was positioned between the ventilator and the intentional leakage port? Considering that the usual reasons for non intentional leaks are "mouth breathing, patient position, phase of the respiratory cycle or respiratory events" as per author description, wouldn't be more appropriate to  positioning the  non intentional leaks port before the intentional leak port?

We thank the reviewer for his/her question regarding the positioning of the unintentional leak. Indeed, since in real life the unintentional leakage comes from the vented port of the masks and the unintentional leaks appears generally from the mouth of around the mask, it would have been more realistic to place the threshold PEP between the intentional leak and the ASL 5000. The reviewer is right. Nevertheless, this does not have any impact on the outcome of our research as both leak ports are placed in series and close to each other.

  1. line 242 please delete the sentence which I guess belongs to the word basic journal format

We thank the reviewer for his sharp eye; it has been deleted.

  1. discussion: among the several options discussed I would suggest to add that to improve intentional leaks, also the careful choice of which interface to use can guarantee the NIV success ref: 10.1080/17476348.2022.2121706

Thank you, a sentence was added into the discussion section and the reference was also linked.

Several strategies can be proposed to reduce leakage such as mask size change, a careful choice of the type of interface”

Reviewer 2 Report

The manuscript assesses the impact of the pattern and intensity of unintentional leakage (UL) using different bilevel ventilators.

Comments:

1.      The bench model consisted of an active artificial lung model (ASL-5000, Ingmar Medical, Pittsburgh, PA, USA) connected to three bilevel ventilators. In the work protocol, you only tested two types of leak patterns, one continuous and one inspiratory only. As the authors of this manuscript acknowledge, it would have been relevant to evaluate a random anarchic pattern of unintentional leakage or expiratory leak. Could you comment a little better on this aspect? If you had done so, what results would you have obtained?

2.      At the end of the discussion, it is indicated that bilevel ventilators do not report leak rate (unintentional and intentional) in the same way among manufacturers, which is detrimental when monitoring NIV. It could not be emphasized more in this aspect, which is crucial for patients undergoing NIV?

Author Response

The manuscript assesses the impact of the pattern and intensity of unintentional leakage (UL) using different bilevel ventilators.

Comments:

  1. The bench model consisted of an active artificial lung model (ASL-5000, Ingmar Medical, Pittsburgh, PA, USA) connected to three bilevel ventilators. In the work protocol, you only tested two types of leak patterns, one continuous and one inspiratory only. As the authors of this manuscript acknowledge, it would have been relevant to evaluate a random anarchic pattern of unintentional leakage or expiratory leak. Could you comment a little better on this aspect? If you had done so, what results would you have obtained?

We thank the reviewer for his/her interesting comment. Indeed, it is likely that the presence of unintended uncontrolled leakage, as in real life, is even more critical. I totally agree with that. Therefore, we added the following sentence in the discussion section:

“[…]we only tested two type of leak patterns, one continuous and one inspiratory only. It would have been relevant to evaluate a random anarchic pattern of unintentional leakage or expiratory leak only, as made by Sogo, Luján et al. (33–35). It is likely that the presence of unintentional random anarchic leakage patterns, can impact even more the functioning of the machines, since the irregular character of the leakage flow is not well corrected by the algorithms. We are currently working on the simulation of random patterns of anarchic leakage, which could allow in the future to evaluate more accurately the behavior of the devices.[…]”

  1. At the end of the discussion, it is indicated that bilevel ventilators do not report leak rate (unintentional and intentional) in the same way among manufacturers, which is detrimental when monitoring NIV. It could not be emphasized more in this aspect, which is crucial for patients undergoing NIV?

Indeed, we decided to enhance the ad-hoc paragraph in the discussion section as follows:

“As with CPAP devices, bilevel ventilators do not report leak rate the same way among manufacturers, which is detrimental when monitoring NIV. For example, some manufacturers report global leaks and others report only estimated unintentional leaks. Some report average leaks, others median leaks, 90th percentile or 95th percentile. For example, some manufacturers report global leaks while others report only estimated unintentional leaks. Some report mean or median leaks, 90th percentile or 95th percentile. This makes it difficult for medical staff to read the devices reports, creates confusion and makes it difficult to change ventilators for the same patient, when necessary, since the leaks data cannot be compared..[…]”

Reviewer 3 Report

The authors conducted a very interesting bench study on a topic that should be considered daily in clinical practice: unintentional leak during non-invasive positive-pressure ventilation. In my opinion, it is very important to conduct such studies and to generate evidence that is then transposed clinically. Moreover, this study is methodologically appropriate in its elaboration and well supports the conclusions.

Below I indicate just a few text formatting changes that need to be made: 

- on line 184: add an affirmative dot after "(Figure2)";

- delete the blank space between lines 186 and 187 that must begin with a capital letter;

- the same applies to line 198 and 199;

- on line 210: please, better clarify the caption of figure #4;

- delete the blank space between line 210 and 2011 that must begin with capital letter;

- the same applies to line 231 and 232.

Author Response

The authors conducted a very interesting bench study on a topic that should be considered daily in clinical practice: unintentional leak during non-invasive positive-pressure ventilation. In my opinion, it is very important to conduct such studies and to generate evidence that is then transposed clinically. Moreover, this study is methodologically appropriate in its elaboration and well supports the conclusions.

Thank you very much for your very positive comment and thorough proofreading

We addressed all the comments below in the main text.

Below I indicate just a few text formatting changes that need to be made: 

- on line 184: add an affirmative dot after "(Figure2)";

- delete the blank space between lines 186 and 187 that must begin with a capital letter;

- the same applies to line 198 and 199;

- on line 210: please, better clarify the caption of figure #4;

Caption of Figure 4 is provided separately with Figure 4 at the bottom of the manuscript.

- delete the blank space between line 210 and 2011 that must begin with capital letter;

- the same applies to line 231 and 232.

Round 2

Reviewer 2 Report

The issues raised in the review have been clarified. Thank you very much